# Dataset of single nucleotide polymorphisms of immune-associated genes in patients with SARS-CoV-2 infection

Nikoletta Katsaouni[1,2,3], Pablo Llavona[4], Yascha Khodamoradi[5], Ann-Kathrin Otto[6], Stephanie Körber[7], Christof Geisen[7], Christian Seidl[7], Maria J. G. T. Vehreschild[5], Sandra Ciesek[8,9,10], Jörg Ackermann[6], Ina Koch[6], Marcel H. Schulz[1,2,3]*, Daniela S. Krause[4,7,11,12,13]*

1 Computational Epigenomics & Systems Cardiology, Institute of Cardiovascular Regeneration, Goethe University and University Clinic, Frankfurt am Main, Germany, 2 German Center for Cardiovascular Research (DZHK), Partner site Rhein Main, Frankfurt am Main, Germany, 3 Cardio-Pulmonary Institute, Goethe University Hospital, Frankfurt am Main, Germany, 4 Georg-Speyer-Haus, Institute for Tumor Biology and Experimental Therapy, Frankfurt am Main, Germany, 5 Department of Internal Medicine, Infectious Diseases, University Hospital Frankfurt, Goethe University Frankfurt, Frankfurt, Germany, 6 Molecular Bioinformatics, Institute of Computer Science, Goethe University Frankfurt am Main, Germany, 7 German Red Cross Blood Donor Service Baden-Württemberg Hessen, Frankfurt am Main, Germany, 8 Institute for Medical Virology, University Hospital, Goethe University, Frankfurt, Germany, 9 German Centre for Infection Research, External Partner Site, Frankfurt, Germany, 10 Fraunhofer Institute for Molecular Biology and Applied Ecology (IME), Branch Translational Medicine and Pharmacology, Frankfurt, Germany, 11 Institute of Biochemistry II and Institute of General Pharmacology and Toxicology, Goethe-University, Frankfurt am Main, Germany, 12 German Cancer Consortium (DKTK), Frankfurt, Germany, 13 Frankfurt Cancer Institute, Frankfurt am Main, Germany

☯ These authors contributed equally to this work.
‡ DSK and MHS also contributed equally to this work.
* Krause@gsh.uni-frankfurt.de (DSK); marcel.schulz@em.uni-frankfurt.de (MHS)

**Data Availability Statement:** The original data files are available in http://www.ncbi.nlm.nih.gov/bioproject/837053 with SRA number: SRP376127 All data needed to each the conclusions drawn in

## Abstract

The SARS-CoV-2 pandemic has affected nations globally leading to illness, death, and economic downturn. Why disease severity, ranging from no symptoms to the requirement for extracorporeal membrane oxygenation, varies between patients is still incompletely understood. Consequently, we aimed at understanding the impact of genetic factors on disease severity in infection with SARS-CoV-2. Here, we provide data on demographics, ABO blood group, human leukocyte antigen (HLA) type, as well as next-generation sequencing data of genes in the natural killer cell receptor family, the renin-angiotensin-aldosterone and kallikrein-kinin systems and others in 159 patients with SARS-CoV-2 infection, stratified into seven categories of disease severity. We provide single-nucleotide polymorphism (SNP) data on the patients and a protein structural analysis as a case study on a SNP in the *SIGLEC7* gene, which was significantly associated with the clinical score. Our data represent a resource for correlation analyses involving genetic factors and disease severity and may help predict outcomes in infections with future SARS-CoV-2 variants and aid vaccine adaptation.

the manuscript and any additional data required to replicate the reported study findings can me found as following: Clinical metadata and HLA types for each patient in the cohort of 159 patients: https://doi.org/10.6084/m9.figshare.21803928.v1. For each participant, Table S1 lists the SNPs and zygosity information (0: homozygous for the reference, 1: heterozygous, 2: homozygous for the alternate). Table S1 is also available online in the open data repository figshare, https://doi.org/10.6084/m9.figshare.20068868.v2. The title of the file is "SNPs per patient and zygosity information."

**Funding:** This work was supported by the Goethe-Corona-Funds of the Goethe University Frankfurt to D.S.K. We acknowledge funding from the Alfons und Gertrud Kassel-Stiftung as part of the center for data science and AI and the DFG Cluster of Excellence Cardio Pulmonary Institute (CPI) [EXC 2026]. We also acknowledge funding from the consortia ACLF-I (Acute Liver Failure - Initiative) and ENABLE (Unraveling mechanisms driving cellular homeostasis, inflammation and infection to enable new approaches in translational medicine) (Hessian Ministry of the Arts and Sciences). The funders had no role in study design, data collection and analysis, decision to publish, or preparation of the manuscript.

**Competing interests:** The authors have declared that no competing interests exist.

## Introduction

The pandemic due to severe acute respiratory syndrome coronavirus 2 (SARS-CoV-2) has devastatingly affected the entire globe causing disease and death to millions, as well as unemployment, social hardship, and shrinkage of global growth, which represents the deepest recession since World War II [1, 2]. While vaccines have become available in industrialized nations and have favorably impacted infection rates, continuously evolving mutations in the viral genome are leading to new waves of infection and causing vaccines to become less efficient [3].

Symptoms of infection with SARS-CoV-2 vary and range from no or mild symptoms, such as headaches, nasal congestion, fever, muscle pain, sore throat, and a loss of taste and smell, to severe illness. In severe cases, dyspnea, hypoxia, and lung involvement, which may lead to respiratory failure, shock, and multi-organ dysfunction, requiring ventilation or extracorporeal membrane oxygenation have been observed [1].

Approximately 80% of patients with SARS-CoV-2 infection recover from the disease without the need for treatment, while approximately 5% are admitted to intensive care, 2.3% require mechanical ventilation and 1.4% of patients die [4]. Advanced age and comorbidities such as diseases of the cardiovascular system, chronic pulmonary diseases, diabetes, cancer, and immunosuppression have been considered risk factors for a severe clinical course or fatal outcome [5]. It is, however, unknown why even certain younger, otherwise healthy individuals or members of certain ethnic groups, may experience severe or even fatal disease or a condition characterized by fatigue and other symptoms termed long COVID.

Since the outbreak of the pandemic, various publications have described polymorphisms in certain genes or variations in expression levels of proteins to influence disease severity and outcome in SARS-CoV-2 infection. Prominent examples are the solute carrier family 6 member 20 (SLC6A20) [1], sialic acid-binding Ig-like lectin 7 (SIGLEC7; CD328) [6], and transmembrane serine protease 2 (TMPRSS2) [7]. SLC6A20 is a member of the family of $Na^+$ and $Cl^-$ coupled transporters expressed by proximal tubule kidney cells (and enterocytes), where they function as transporters of proline [8, 9]. SLC6A20 has been identified as part of a 3p21.31 gene cluster representing a genetic susceptibility locus in SARS-CoV-2 patients with respiratory failure [1]. By hetero-dimerization with an angiotensin-converting enzyme (ACE)2, SLC6A20 is associated with the renin-angiotensin system. In the renin-angiotensin system, ACE2 plays a key role in converting the vasoconstrictor angiotensin-II (Ang-II) into the vasodilator Ang1-7 and, thereby, lowering blood pressure [10]. TMPRSS2 promotes the uptake of SARS-CoV-2 via proteolytic cleavage of the ACE2 receptor. Single nucleotide polymorphisms in TMPRSS2 have been shown to vary amongst global populations [7] and to affect the susceptibility of a patient to infection with SARS-CoV-2 via modulation of splicing, miRNA expression, etc. [11]. SIGLEC7 facilitates the inhibition of the cytotoxicity of natural killer cells [12]. Expression of SIGLEC7 has been shown to correlate with SARS-CoV-2 levels in nasopharyngeal swabs or autopsies [6].

Based on these previous findings, we collected demographics, ABO blood group, human leukocyte antigen (HLA) type, data on hereditary prothrombotic factors, as well as next-generation sequencing data of genes in the natural killer cell receptor family, the renin-angiotensin-aldosterone and kallikrein-kinin systems in patients. Patients had been stratified into seven categories of disease severity according to the World Health Organization (WHO) Ordinal Scale for Clinical Improvement. We hypothesized that ABO blood group, HLA type, hereditary factors or certain SNPs in immune-, blood pressure- or inflammation-associated genes or a combination of these factors may impact severity of symptoms and influence outcome in infection with SARS-CoV-2. We, therefore, collected blood samples from 159 adult individuals with past or current infection with SARS-CoV-2, in order to test for factors, possibly predisposing them to severe outcome.

Taken together, our data may aid predictions on the susceptibility of individuals to novel virus variants, their anticipated disease severity, as well as on the efficiency of current and future vaccines to viral mutants. It is hoped that our data will support the rational design, adaptation, and optimization of vaccines to new viral strains.

## Materials and methods

### Patient consent

This study was performed as part of the CAP-Net Foundation's competence network on community-acquired pneumonia and was conducted according to the guidelines of the Declaration of Helsinki and international standards of good clinical practice. The ethics committee of the Goethe University Frankfurt approved the protocol and any protocol amendments (Protocol number: 20–748). All enrolled patients provided a written informed consent form and specifically agreed to the performance of genetic studies, the transfer of their anonymized data to third parties and the use of data from their medical records in this research. All data were fully anonymized prior to their access. No minors were included in the study.

### Patient collection

10 ml of peripheral blood were collected in EDTA from 159 patients, who had been or were currently infected with SARS-CoV-2 and who received medical care from the university hospital of the Goethe-University in Frankfurt am Main, Germany. All patient samples were collected between April 2020 and November 2021, a period when the virus strains B.1.1.7 and B.1.617 or B.1.617.2 were most prevalent. All included patients tested positive by polymerase chain reaction (PCR) at different institutions. DNA for HLA typing was available for 156 of the patients. 67 patients were female (42%), 90 patients were male (56%), and two patients were not reported. SNP sequencing was conducted for all patients. The patients were between 18–92 years of age. The patients were stratified according to the World Health Organization Ordinal Scale for Clinical Improvement [13], with a

   score 1 signifying very mild symptoms without compromise of activities,
   score 2 signifying mild symptoms with a compromise of activities,
   score 3 signifying moderate disease requiring admission to hospital (but no oxygen),
   score 4 signifying hospital care with oxygen treatment,
   score 5 signifying severe disease with high oxygen requirements,
   score 6 signifying intubation and mechanical ventilation and
   score 7 signifying ventilation plus the support of organs by pressors or extracorporeal membrane oxygenation (ECMO).

The scoring system for these patients was also compared to an updated scoring system [14]. For further data and future statistical analysis, patients were categorized as having mild ($\leq 2$, n = 94) or severe disease ($\geq 3$, n = 65), with the categorization 'severe disease' defining patients who were hospitalized. Respiratory hospitalization was defined as hospitalization due to dyspnea requiring oxygen treatment, mechanical ventilation, or ECMO (scores $\geq 4$, n = 51). The steps of the experimental workflow can be found in Fig 1.

Table 1 (Excel version online available at https://doi.org/10.6084/m9.figshare.21803928.v1) lists the gender, age, clinical score, ABO type, Rhesus (Rh) type, HLA types, ethnicity, and clinical information on comorbidities for each patient. Information on the classification of the patients according to the WHO Clinical Improvement clinical progression scale [14] is provided and can be found in the column "Clinical Score". Note that sample IDs 33, 45, and 79 are not included.

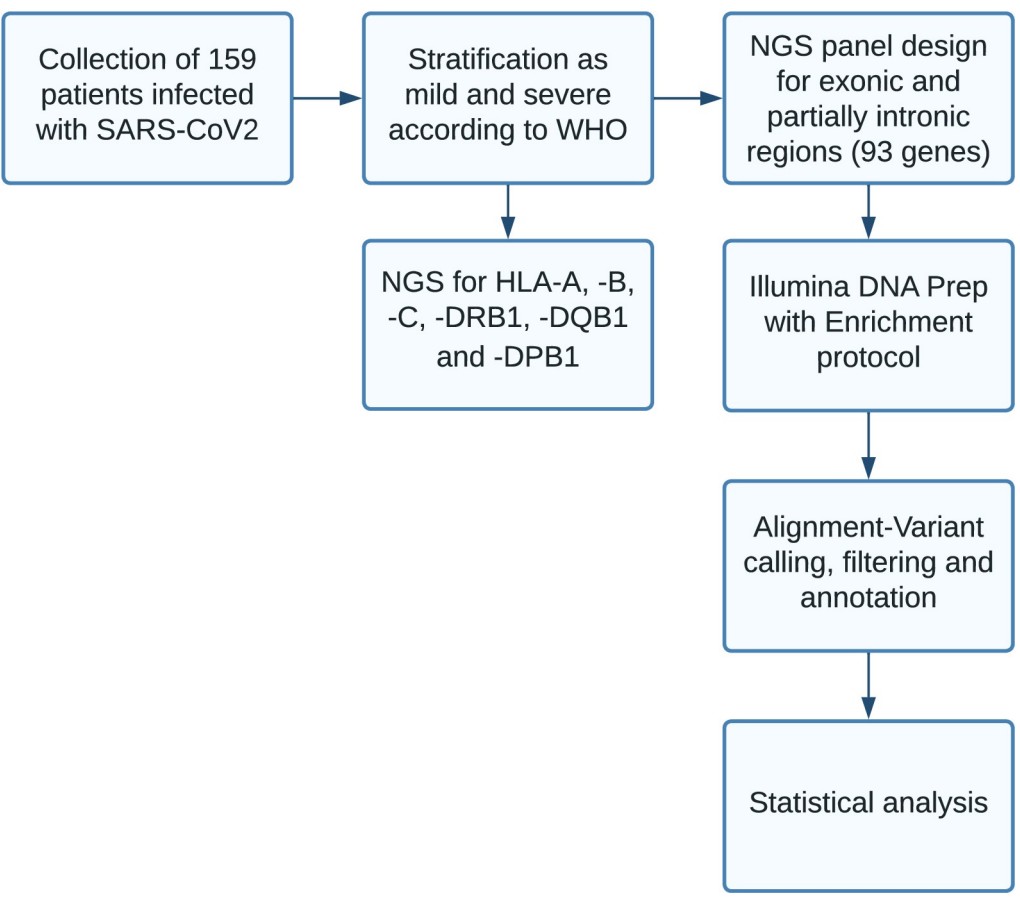

**Fig 1. Experimental workflow.**

## Blood typing

Blood typing for A, B, and O was performed using the Bioclone Kit from Ortho-Clinical Diagnostics (Unterschleissheim, Germany) according to the manufacturer's instructions. Rh was determined using the IgM Anti-D Mono-Type reagent from Medion Grifols Diagnostic AG (Düdingen, Switzerland).

## DNA isolation

Red blood cells (RBCs) were lysed with ACK lysis buffer (Thermo Fisher, Waltham, MA). The remaining cells were washed and centrifuged (at 1780 x g for 15 min), before they were collected in resuspension buffer (NaCl 6M, $Na_2$EDTA pH 8), to which proteinase K (10 mg/ml) and 20% w/v sodium dodecyl sulfate (SDS) solution for overnight cell lysis at 37°C had been added. Cell debris was separated from nucleic acids after incubation in 6 M NaCl at 55°C for 15 min. The DNA was purified with 96% and 70% ethanol and, finally, resuspended in TE buffer.

## HLA genotyping

Next-generation sequencing for HLA-A, -B, -C, -DRB1, -DQB1, and -DPB1 was performed by a high-resolution amplicon-based approach using NGSgo-MX6- and NGSgo-LibrX kits (GENDX, Utrecht, The Netherlands) according to the manufacturer's instructions on an Illumina Mini-Seq device (Illumina, San Diego, CA).

**Table 1. Clinical metadata and HLA types for each individual patient in the cohort of 159 patients.** Note that, sample IDs 33, 45, and 79 are not included. Excel version is online available at https://doi.org/10.6084/m9.figshare.21803928.v1.

| Sample | Sex | Age | Clinical Score | Revised Clinical Score | ABO Type | Rhesus | HLA-A | HLA-B | HLA-C | HLA-DRB1 | HLA-DQB1 | HLA-DPB1 | FVL | Prothrombin | PCR test | Ethnicity | Premorbidities |
|---|---|---|---|---|---|---|---|---|---|---|---|---|---|---|---|---|---|
| 1 | f | 47 | 2 | 2 | O | pos | 3; 30 | 35;51 | 4; 14 | 1;7 | 2;5 | 4;17 | WT | WT | pos | Caucasian | |
| 2 | m | 51 | 2 | 2 | O | pos | 2; 3 | 7;44 | 5;7 | 4;15 | 3;6 | 4;14 | WT | WT | pos | Caucasian | |
| 3 | f | 34 | 2 | 2 | O | pos | 25; 32 | 14;18 | 8;12 | 7;14 | 2;5 | 4;10 | WT | WT | pos | Caucasian | |
| 4 | f | 52 | 2 | 2 | O | pos | 2; | 15;40 | 3;3 | 11;13 | 3;6 | 3;4 | het | WT | pos | Caucasian | |
| 5 | f | 27 | 2 | 2 | A | pos | 1; | 8;37 | 6;7 | 3;10 | 2;5 | 4;4 | WT | WT | pos | Caucasian | |
| 6 | m | 67 | 2 | 2 | A | pos | 2; 11 | 27;51 | 2;15 | 4;9 | 3;3 | 4;4 | WT | WT | pos | Caucasian | |
| 7 | m | 78 | 6 | 10 | B | pos | 1; 25 | 8;18 | 7;12 | 3;4 | 2;3 | 4;4 | WT | WT | pos | Caucasian | |
| 8 | f | 71 | 6 | 9 | A | pos | 11; 26 | 35; | 4; | 14;15 | 5;6 | 1;4 | WT | WT | pos | Caucasian | |
| 9 | m | 52 | 7 | 7 | A | pos | 1; 2 | 15;51 | 1;7 | 11;13 | 3;6 | 124;4 | WT | WT | pos | Middle Eastern | |
| 10 | m | 61 | 6 | 10 | A | pos | 2; 3 | 7;51 | 2;7 | 11;15 | 3;6 | 2;4 | hom | WT | pos | Caucasian | |
| 11 | m | 37 | 2 | 2 | O | pos | 2; 26 | 38;40 | 3;12 | 13;13 | 3;6 | 4;4 | | | pos | Caucasian | |
| 12 | m | 32 | 4 | 4 | O | pos | 2;3 | 42;57 | 17;18 | 3;13 | 4;6 | 1;2 | WT | WT | pos | Hispanic | |
| 13 | f | 68 | 4 | 5 | O | neg | 32;69 | 7;38 | 7;12 | 15;15 | 6;6 | 4;4 | WT | WT | pos | Caucasian | Hypothyroidism |
| 14 | f | 40 | 1 | 1 | A | neg | 2; 30 | 7; 13 | 6;7 | 7;11 | 2;3 | 4;14 | WT | WT | pos | North African | Penicillin Allergy |
| 15 | m | 47 | 2 | 3 | O | pos | 2; 24 | 15;51 | 3; 14 | 13;13 | 6;6 | 4;4 | | | pos | Caucasian | |
| 16 | m | 46 | 2 | 2 | A | pos | 2; 11 | 8;27 | 1;7 | 4;15 | 3;6 | 2;14 | WT | WT | pos | Caucasian | Ulcerative colitis |
| 17 | m | 41 | 1 | 2 | O | pos | 2; 11 | 7;7 | 7;7 | 4;15 | 3;6 | 4;4 | WT | | pos | Caucasian | |
| 18 | f | 37 | 2 | 3 | A | pos | 2; 3 | 15;37 | 3;6 | 4;10 | 3;5 | 3;4 | WT | WT | pos | Caucasian | |
| 19 | m | 67 | 4 | 4 | AB | pos | 2;31 | 41;51 | 15;17 | 11; | 3 | 4;4 | het | WT | pos | Caucasian | |
| 20 | m | 50 | 4 | 5 | AB | pos | 2; 24 | 51;55 | 3; 16 | 4;16 | 3;5 | 1;416 | WT | WT | pos | North African | |
| 21 | f | 22 | 2 | 2 | B | pos | 3; 26 | 27;56 | 1; 1 | 15;16 | 5;6 | 4;10 | WT | WT | pos | Caucasian | |
| 22 | f | 35 | 2 | 2 | O | pos | 2; 24 | 51;51 | 1; 2 | 11; | 3;3 | 4;4 | WT | WT | pos | Caucasian | |
| 23 | m | 27 | 2 | 2 | A | pos | 2; 2 | 15;51 | 3; 4 | 3;4 | 2;3 | 4;4 | WT | WT | pos | Caucasian | |
| 24 | m | 50 | 4 | 8 | A | pos | 2; 3 | 15;41 | 14;17 | 3;4 | 2;3 | 3;650 | het | WT | pos | Caucasian | |
| 25 | m | 65 | 4 | 10 | A | pos | 2;29 | 7;40 | 3;15 | 1;10 | 5 | 4;10 | WT | WT | pos | Caucasian | |
| 26 | f | 64 | 2 | 3 | O | pos | 2; 24 | 7;44 | 4;7 | 7;15 | 2;6 | 4;14 | WT | WT | pos | Caucasian | |
| 27 | f | 66 | 2 | 2 | O | pos | 1; 3 | 7;35 | 4;7 | 7;15 | 2;6 | 4;4 | WT | WT | pos | Caucasian | |
| 28 | f | 64 | 4 | 5 | O | pos | 1; 3 | 7;35 | 4;7 | 7;15 | 2;6 | 4;4 | WT | WT | pos | Caucasian | |
| 29 | m | 65 | 2 | 2 | O | pos | 2; 2 | 15;51 | 1; 3 | 1;4 | 3;5 | 2;4 | WT | WT | pos | Caucasian | COPD |
| 30 | f | 63 | 1 | 2 | O | pos | 2; 3 | 15;18 | 3;12 | 1;11 | 3;5 | 1;4 | WT | WT | pos | Caucasian | |
| 31 | f | 45 | 3 | 4 | B | pos | 3; 32 | 7;39 | 7;12 | 8;15 | 4;5 | 2;4 | WT | WT | pos | Caucasian | Atopic dermatitis |
| 32 | m | 67 | 2 | 2 | O | pos | 2; 3 | 39;44 | 5;7 | 7;14 | 2;5 | 4;4 | WT | WT | pos | Caucasian | Arthritis, Hashimoto"s Thyroiditis |
| 34 | m | 49 | 1 | 1 | O | pos | 1;32 | 8;38 | 7;12 | 14;15 | 5;6 | 4;4 | WT | WT | pos | Caucasian | Coronary artery disease |

(Continued)

**Table 1.** (Continued)

| | | | | | | | | | | | | | | | | | |
|---|---|---|---|---|---|---|---|---|---|---|---|---|---|---|---|---|---|
| 35 | m | 39 | 3 | 4 | O | pos | 11; 11 | 15; 55 | 3; 4 | 12; 12 | 3; 3 | 5; 5 | WT | WT | pos | East Asian | |
| 36 | m | 70 | 6 | 7 | B | neg | 3; 23 | 7; 44 | 4; 7 | 15; 15 | 6; 6 | 3; 4 | WT | WT | pos | Caucasian | |
| 37 | m | 63 | 7 | 7 | A | pos | 2; 11 | 7; 7 | 7; 7 | 4; 15 | 3; 6 | 4; 4 | WT | WT | pos | Middle Eastern | |
| 38 | m | 42 | 2 | 3 | AB | pos | 1; 32 | 35; 57 | 4; 6 | 7; 13 | 3; 5 | 4; 4 | WT | het | pos | Caucasian | |
| 39 | f | 47 | 2 | 2 | A | neg | 3; 25 | 7; 13 | 6; 7 | 3; 7 | 2; 2 | 4; 4 | WT | WT | pos | Caucasian | |
| 40 | m | 56 | 2 | 2 | O | neg | 2; 3 | 15; 40 | 3; 3 | 4; 7 | 2; 3 | 6; 15 | WT | WT | pos | Caucasian | |
| 41 | f | 32 | 2 | 2 | A | neg | 2; 68 | 14; 27 | 2; 8 | 13; 15 | 3; 6 | 2; 19 | WT | WT | pos | Caucasian | |
| 42 | m | 52 | 2 | 2 | B | neg | 1; 24 | 8; 44 | 4; 7 | 3; 11 | 2; 3 | 1; 17 | WT | WT | pos | Middle Eastern | |
| 43 | f | 50 | 3 | 5 | B | pos | 1; 2 | 8; 57 | 6; 7 | 3; 15 | 2; 6 | 4; 13 | WT | WT | pos | Caucasian | Meningioma, chronically relapsing EBV infection, asthma, bone cyst left wrist, thyroid nodules |
| 44 | f | 32 | 2 | 2 | O | pos | 2; 2 | 44; 51 | 4; 4 | 7; 16 | 2; 5 | 13; 14 | WT | WT | pos | Caucasian | |
| 46 | m | 89 | 4 | 6 | B | pos | 11; 33 | 14; 38 | 8; 12 | 1; 3 | 2; 5 | 1; 2 | WT | WT | pos | Caucasian | Hypertension, atrial flutter |
| 47 | f | 51 | 2 | 2 | A | pos | 2; 3 | 15; 51 | 2; 6 | 13; 13 | 6; 6 | 4; 9 | WT | WT | pos | Caucasian | Asthma |
| 48 | m | 55 | 2 | 2 | O | pos | 2; 24 | 7; 38 | 7; 12 | 15; 15 | 6; 6 | 2; 10 | WT | WT | pos | Caucasian | |
| 49 | m | 54 | 5 | 7 | O | pos | 11; 68 | 53; 55 | 3; 4 | 1; 13 | 5; 6 | 4; 10 | WT | WT | pos | Caucasian | Hypertension, arthritis, thrombosis (Microthrombosis in the fingers) |
| 50 | f | 28 | 2 | 2 | A | pos | 2; 24 | 40; 51 | 14; 15 | 8; 11 | 3; 3 | 4; 16 | WT | WT | pos | East Asian | Systemic lupus erythematodes |
| 51 | m | 30 | 2 | 2 | A | pos | 2; 11 | 35; 56 | 4; 4 | 8; 15 | 4; 6 | 13; 26 | WT | WT | pos | East Asian | Hypothyroidism |
| 52 | m | 56 | 4 | 5 | A | pos | 3; 24 | 7; 7 | 7; 7 | 4; 15 | 3; 6 | 3; 4 | WT | WT | pos | Caucasian | |
| 53 | m | 56 | 2 | 2 | A | pos | 11; 24 | 40; 51 | 3; 15 | 4; 11 | 3; 3 | 3; 4 | WT | WT | pos | Caucasian | Hypertension |
| 54 | m | 38 | 2 | 2 | A | pos | 3; 11 | 35; 35 | 4; 5 | 7; 11 | 2; 3 | 3; 4 | WT | WT | pos | Caucasian | Hypertonie |
| 55 | f | 54 | 2 | 2 | O | pos | 1; 2 | 8; 39 | 7; 7 | 3; 8 | 2; 4 | 3; 4 | WT | WT | pos | Caucasian | |
| 56 | m | 51 | 2 | 2 | O | pos | 2; 25 | 8; 18 | 7; 12 | 3; 13 | 2; 6 | 4; 14 | WT | WT | pos | Caucasian | |
| 57 | m | 42 | 2 | 2 | A | pos | 1; 3 | 8; 15 | 3; 7 | 3; 4 | 2; 3 | 1; 10 | WT | WT | pos | Caucasian | |
| 58 | f | 29 | 1 | 2 | B | pos | 2; 32 | 27; 57 | 1; 6 | 3; 14 | 2; 5 | 4; 5 | WT | WT | pos | Caucasian | (Pregnant) |
| 59 | m | 59 | 4 | 5 | O | neg | 29; 68 | 8; 45 | 6; 7 | 3; 4 | 2; 3 | 4; 10 | WT | WT | pos | Caucasian | Asthma |
| 60 | | | 2 | 2 | A | pos | 2; 25 | 14; 18 | 8; 12 | 1; 16 | 5; 5 | 3; 4 | WT | WT | pos | Caucasian | |
| 61 | f | 42 | 2 | 2 | O | pos | 2; 3 | 18; 38 | 7; 12 | 3; 15 | 2; 5 | 2; 126 | WT | WT | pos | Hispanic | Hyperthyroidism |
| 62 | m | 49 | 2 | 2 | O | pos | 1; 2 | 15; 51 | 7; 16 | 4; 7 | 3; 3 | 4; 4 | WT | WT | pos | Caucasian | Essential thrombocythemia |
| 63 | f | 28 | 2 | 2 | O | pos | 2; 2 | 15; 44 | 4; 16 | 4; 7 | 2; 3 | 1; 15 | WT | WT | pos | Caucasian | |
| 64 | m | 30 | 3 | 3 | O | neg | 2; 68 | 7; 53 | 4; 7 | 13; 15 | 6; 6 | 4; 4 | WT | WT | pos | Caucasian | |
| 65 | m | 30 | 2 | 2 | O | pos | 1; 24 | 27; 57 | 5; 6 | 7; 11 | 3; 3 | 4; 4 | WT | WT | pos | Caucasian | |
| 66 | m | 18 | 2 | 2 | O | neg | 3; 32 | 7; 7 | 7; 7 | 8; 15 | 4; 6 | 4; 4 | WT | WT | pos | Caucasian | |
| 67 | m | 29 | 2 | 2 | O | neg | 3; 3 | 7; 7 | 7; 7 | 15; 15 | 6; 6 | 3; 4 | WT | WT | pos | Caucasian | Lymphoma 2014 |
| 68 | f | 39 | 2 | 2 | O | pos | 1; 24 | 14; 27 | 6; 7 | 3; 7 | 2; 2 | 1; 17 | WT | WT | pos | Caucasian | Psoriasis, allergic rhinitis |
| 69 | f | 58 | 2 | 2 | A | pos | 2; 2 | 40; 45 | 3; 6 | 10; 13 | 5; 6 | 3; 3 | WT | WT | pos | Caucasian | |
| 70 | m | 36 | 2 | 2 | A | neg | 1; 32 | 8; 14 | 7; 8 | 3; 7 | 2; 2 | 4; 4 | WT | WT | pos | Caucasian | |
| 71 | m | 35 | 2 | 2 | O | pos | 2; 68 | 40; 44 | 1; 5 | 1; 4 | 3; 5 | 5; 20 | WT | WT | pos | Caucasian | Diabetes mellitus II, Hypertension |
| 72 | m | 46 | 2 | 2 | A | pos | 1; 3 | 7; 15 | 7; 7 | 4; 12 | 3; 3 | 4; 4 | WT | WT | pos | Caucasian | none |

(Continued)

**Table 1.** (Continued)

| # | Sex | Age | | | ABO | | | | | | | | | | | | Ethnicity | Disease |
|---|---|---|---|---|---|---|---|---|---|---|---|---|---|---|---|---|---|---|
| 73 | f | 33 | 1 | 2 | A | pos | 3; 68 | 7; 44 | 7; 7 | 7; 13 | 2; 6 | 350; 905 | WT | WT | WT | pos | Caucasian | Allergy (corn + pollen); Hypertension |
| 74 | f | 60 | 2 | 2 | O | pos | | | | | | | WT | WT | WT | pos | Caucasian | allergic asthma |
| 75 | m | 46 | 2 | 4 | A | pos | 2; 24 | 18; 50 | 6; 12 | 7; 11 | 2; 3 | 4; 9 | WT | WT | WT | pos | Caucasian | |
| 76 | f | 50 | 1 | 2 | A | neg | 1; 68 | 8; 53 | 4; 7 | 3; 13 | 2;6 | 104; 4 | het | WT | WT | pos | Caucasian | Lung carcinoma |
| 77 | f | 51 | 2 | 2 | O | pos | 24; 26 | 7; 27 | 2; 7 | 15; 15 | 6; 6 | 3; 4 | WT | WT | WT | pos | Caucasian | Rheumatoid arthritis |
| 78 | m | 55 | 2 | 2 | O | pos | 1; 24 | 7; 51 | 7; 14 | 13; 15 | 6; 6 | 2; 2 | WT | WT | WT | pos | Caucasian | |
| 80 | f | 26 | 2 | 2 | A | pos | 1; 30 | 8; 13 | 6; 7 | 3; 7 | 2; 2 | 4; 4 | WT | WT | WT | pos | Caucasian | Allergies (Cat; pollen) |
| 81 | m | 59 | 2 | 2 | A | pos | 23; 24 | 15; 56 | 3; 4 | 3; 7 | 2; 2 | 13; 13 | WT | WT | WT | pos | Caucasian | |
| 82 | f | 32 | 1 | 2 | A | pos | 1; 3 | 7; 55 | 3; 7 | 11; 15 | 3; 6 | 3; 4 | WT | WT | WT | pos | Caucasian | |
| 83 | f | 40 | 2 | 2 | B | pos | 2; 2 | 15; 44 | 3; 16 | 4; 14 | 3; 5 | 2; 2 | WT | WT | WT | pos | Caucasian | Hemophilia A, autoimmune thyroid disease |
| 84 | m | 37 | 2 | 2 | AB | pos | 2; 32 | 44; 50 | 5; 6 | 7; 11 | 2; 3 | 2; 4 | WT | WT | WT | pos | Caucasian | |
| 85 | m | 30 | 2 | 2 | B | pos | 3; 29 | 7; 35 | 4; 15 | 11; 13 | 3; 6 | 4; 17 | WT | WT | WT | pos | Caucasian | Allergy (Grasses), Hailey Hailey disease |
| 86 | f | 39 | 2 | 2 | A | pos | 2; 31 | 51; 51 | 15; 15 | 11; 13 | 3; 6 | 2; 2 | WT | WT | WT | pos | Caucasian | |
| 87 | f | 41 | 2 | 2 | B | pos | 1; 30 | 18; 37 | 5; 6 | 3; 10 | 2; 5 | 2; 2 | WT | WT | WT | pos | Caucasian | Malignant melanoma |
| 88 | f | 22 | 2 | 2 | O | pos | 2; 3 | 7; 14 | 7; 8 | 13; 15 | 3; 5 | 2; 2 | WT | WT | WT | pos | East Asian | Asthma |
| 89 | m | 53 | 2 | 2 | A | pos | | | | 13; | | | | WT | WT | pos | Caucasian | |
| 90 | m | 50 | 2 | 2 | O | pos | 3; 32 | 7; 7 | 7; 7 | 8; 15 | 4; 6 | 3; 16 | WT | WT | WT | pos | Caucasian | |
| 91 | m | 25 | 2 | 2 | O | pos | 3; 3 | 7; 57 | 6; 7 | 15; 15 | 6; 6 | 3; 4 | WT | WT | WT | pos | Caucasian | Diabetes mellitus II |
| 92 | m | 57 | 2 | 2 | A | pos | 1; 68 | 44; 44 | 4; 5 | 4; 7 | 2; 3 | 4; 4 | WT | WT | WT | pos | Caucasian | Atopic dermatitis, allergic asthma |
| 93 | m | 35 | 2 | 2 | B | pos | 1; 2 | 13; 15 | 3; 6 | 4; 7 | 2; 3 | 4; 17 | WT | WT | WT | pos | Caucasian | Hypertension |
| 94 | f | 34 | 2 | 2 | B | pos | 11; 33 | 14; 35 | 4; 8 | 1; 1 | 5; 5 | 2; 104 | WT | WT | WT | pos | Caucasian | |
| 95 | m | 41 | 2 | 2 | A | pos | 2; 25 | 40; 44 | 3; 4 | 4; 10 | 3; 5 | 3; 4 | WT | WT | WT | pos | Caucasian | |
| 96 | f | 42 | 4 | 5 | O | pos | 3; 26 | 38; 51 | 1; 12 | 13; | 3; 6 | 4; 5 | WT | WT | WT | pos | Caucasian | |
| 97 | f | 29 | 2 | 2 | B | pos | 25; 31 | 40; 44 | 3; 5 | 4; 15 | 3; 6 | 1; 3 | WT | WT | WT | pos | Caucasian | Epilepsy |
| 98 | f | 27 | 2 | 2 | O | pos | 1; 11 | 7; 40 | 3; 7 | 4; 15 | 3; 6 | 1; 4 | WT | WT | WT | pos | Caucasian | Hashimoto's Thyroiditis, slipped disc |
| 99 | f | 49 | 2 | 2 | O | pos | 23; 24 | 49; 49 | 7; 7 | 1; 11 | 3; 5 | 104; 4 | WT | WT | WT | pos | Caucasian | |
| 100 | f | 25 | 1 | 2 | B | pos | 2; 23 | 49; 49 | 7; 7 | 4; 11 | 3; 3 | 4; 13 | WT | WT | WT | pos | Caucasian | Hashimoto's Thyroiditis |
| 101 | m | 20 | 3 | 4 | O | pos | 1; 25 | 18; 40 | 2; 5 | 3; 11 | 2; 3 | 4; 4 | WT | WT | WT | pos | Caucasian | |
| 102 | f | 21 | 2 | 2 | A | pos | 2; 24 | 35; 40 | 3; 4 | 11; 13 | 3; 6 | 4; 4 | WT | WT | WT | pos | Caucasian | |
| 103 | f | 51 | 2 | 2 | A | pos | 2; 3 | 40; 40 | 3; 3 | 4; 7 | 2; 3 | 4; 4 | WT | WT | WT | pos | Caucasian | Iron deficiency anemia |
| 104 | m | 48 | 2 | 2 | O | pos | 2; 2 | 7; 44 | 5; 7 | 13; 15 | 6; 6 | 4; 4 | het | het | WT | pos | Caucasian | |
| 105 | f | 49 | 2 | 2 | A | pos | 2; 68 | 8; 53 | 4; 7 | 12; 13 | 3; 6 | 2; 4 | het | het | WT | pos | Caucasian | |
| 106 | m | 30 | 1 | 2 | A | pos | 2; 2 | 13; 51 | 6; 15 | 7; 13 | 2; 6 | 4; 17 | WT | WT | WT | pos | Caucasian | Deficiency of factor VII |
| 107 | f | 43 | 2 | 2 | B | pos | 2; 30 | 13; 18 | 5; 6 | 3; 10 | 2; 5 | 3; 4 | WT | WT | WT | pos | Caucasian | |
| 108 | m | 58 | 2 | 2 | A | pos | 1; 3 | 7; 8 | 7; 7 | 9; 15 | 3; 6 | 2; 4 | WT | WT | WT | pos | Caucasian | |
| 109 | m | 56 | 5 | 6 | O | pos | 2; 24 | 15; 51 | 3; 15 | 13; 15 | 6; 6 | 2; 2 | WT | WT | WT | pos | Caucasian | |
| 110 | f | 56 | 2 | 2 | A | pos | 2; 68 | 14; 51 | 8; 15 | 8; 13 | 3; 4 | 2; 4 | WT | WT | WT | pos | Caucasian | |
| 111 | f | 50 | 2 | 2 | O | pos | 2; 2 | 7; 44 | 5; 7 | 4; 11 | 3; 3 | 1; 4 | WT | WT | WT | pos | Caucasian | |
| 112 | m | 49 | 2 | 2 | A | pos | 11; 30 | 18; 51 | 5; 15 | 3; 3 | 2; 2 | 4; 13 | WT | WT | WT | pos | Caucasian | Migraine; Endometriosis |

(*Continued*)

**Table 1.** (Continued)

| ID | sex | age | | | | | | | | | | | | | | | |
|----|-----|-----|---|---|---|-----|------|-------|------|------|-----|-------|-----|-----|-----|----------------|------------------------------|
| 113 | f | 49 | 2 | 2 | O | pos | 1; 26 | 35; 44 | 3; 4 | 1; 7 | 2; 5 | 2; 17 | WT | WT | pos | Caucasian | Hashimoto's Thyroiditis |
| 114 | m | 36 | 2 | 2 | B | pos | 3; 26 | 7; 27 | 1; 7 | 1; 15 | 5; 6 | 2; 3 | WT | WT | pos | Caucasian | Thyroid nodules |
| 115 | f | 49 | 3 | 4 | A | neg | 11; 29 | 45; 58 | 3; 6 | 13; 15 | 6; 6 | 2; 3 | WT | WT | pos | Caucasian | Hashimoto's Thyroiditis |
| 116 | f | 27 | 2 | 2 | O | pos | 2; 3 | 38; 50 | 6; 12 | 7; 7 | 2; 3 | 4; 4 | WT | WT | pos | Caucasian | Migraine |
| 117 | f | 41 | 2 | 2 | O | pos | 2; 11 | 44; 51 | 7; 15 | 4; 11 | 3; 3 | 4; 15 | WT | WT | pos | Caucasian | |
| 118 | m | 55 | 3 | 4 | O | pos | 2; 68 | 44; 44 | 4; 5 | 13; 13 | 6; 6 | 4; 14 | WT | WT | pos | Caucasian | |
| 119 | f | 40 | 2 | 2 | O | pos | 2; 30 | 14; 18 | 5; 8 | 7; 11 | 2; 3 | 4; 4 | WT | WT | pos | Caucasian | |
| 120 | f | 22 | 2 | 2 | A | pos | 68; 68 | 13; 50 | 6; 16 | 7; 13 | 2; 3 | 104; 4 | WT | WT | pos | Caucasian | Allergic asthma, breast cancer |
| 121 | f | 51 | 2 | 2 | A | pos | 2; 68 | 7; 44 | 5; 7 | 7; 15 | 2; 6 | 2; 4 | WT | WT | pos | Black | |
| 122 | f | 33 | 2 | 2 | A | pos | 1; 2 | 13; 18 | 6; 7 | 7; 11 | 2; 3 | 3; 4 | WT | WT | pos | Caucasian | Psoriasis, gout, hip dysplasia |
| 123 | f | 75 | 4 | 5 | O | pos | 3; 3 | 7; 35 | 4; 7 | 1; 15 | 5; 6 | 4; 4 | WT | WT | pos | Caucasian | Myokarditis, perikarditis |
| 124 | m | 64 | 4 | 5 | A | pos | 2; 32 | 35; 44 | 4; 5 | 15; 15 | 6; 6 | 2; 2 | het | het | pos | Caucasian | |
| 125 | m | 60 | 4 | 5 | O | pos | 2; 23 | 8; 44 | 2; 7 | 3; 7 | 2; 2 | 4; 835 | WT | WT | pos | Caucasian | Hashimoto's Thyroiditis |
| 126 | m | 81 | 7 | 9 | O | pos | 1; 3 | 15; 51 | 7; 14 | 1; 13 | 5; 6 | 2; 4 | WT | WT | pos | Middle Eastern | |
| 127 | m | 56 | 5 | 6 | O | pos | 24; 24 | 35; 51 | 04; 16 | 8; 13 | 3; 6 | 2; 4 | WT | WT | pos | North African | |
| 128 | f | 52 | 3 | 4 | O | pos | 1; 3 | 7; 7 | 7; 7 | 12; 15 | 3; 6 | 4; 4 | WT | WT | pos | Middle Eastern | |
| 129 | m | 37 | 5 | 6 | A | pos | 11; 32 | 35; 40 | 4; 15 | 4; 7 | 3; 3 | 4; 4 | WT | WT | pos | Caucasian | |
| 130 | m | 67 | 4 | 5 | A | pos | 2; 68 | 44; 52 | 7; 12 | 11; 15 | 3; 6 | 4; 4 | WT | WT | pos | Middle Eastern | |
| 131 | m | 34 | 4 | 5 | O | pos | 2; 30 | 39; 53 | 4; 7 | 4; 13 | 2; 6 | 104; 4 | WT | WT | pos | Middle Eastern | |
| 132 | m | 82 | 4 | 5 | O | pos | 11; 24 | 35; 52 | 2; 12 | 11; 15 | 3; 6 | 3; 4 | WT | WT | pos | Black | |
| 133 | m | 34 | 3 | 4 | A | pos | 11; 11 | 52; 52 | 12; 12 | 15; 15 | 6; 6 | 2; 2 | WT | WT | pos | North African | |
| 134 | f | 41 | 4 | 5 | A | pos | 29; 30 | 53; 58 | 4; 7 | 1; 3 | 4; 5 | 1; 104 | WT | WT | pos | Caucasian | |
| 135 | f | 80 | 4 | 5 | A | pos | 2; 3 | 7; 35 | 2; 7 | 12; 15 | 3; 6 | 4; 4 | WT | WT | pos | North African | |
| 136 | m | 41 | 3 | 4 | AB | pos | 1; 32 | 35; 57 | 4; 6 | 7; 15 | 3; 6 | 4; 4 | WT | WT | pos | Caucasian | |
| 137 | m | 49 | 3 | 4 | O | pos | 29; 29 | 15; 49 | 3; 7 | 12; 15 | 5; 6 | 2; 665 | WT | WT | pos | East Asian | |
| 138 | m | 52 | 4 | 5 | A | pos | 2; 3 | 7; 35 | 4; 7 | 3; 16 | 2; 5 | 4; 4 | WT | WT | pos | Black | |
| 139 | m | 65 | 4 | 5 | A | pos | 2; 24 | 38; 49 | 7; 12 | 11; 13 | 3; 6 | 2; 4 | WT | WT | pos | Caucasian | |
| 140 | m | 82 | 4 | 5 | A | pos | 2; 2 | 15; 39 | 2; 3 | 1; 1 | 5; 5 | 4; 4 | WT | WT | pos | North African | |
| 141 | m | 65 | 5 | 6 | A | pos | 11; 24 | 39; 51 | 7; 16 | 1; 13 | 5; 6 | 4; 10 | WT | WT | pos | Caucasian | |
| 142 | m | 59 | 4 | 5 | A | pos | 2; 33 | 8; 53 | 4; 7 | 3; 13 | 2; 6 | 4; 4 | WT | WT | pos | Caucasian | |
| 143 | m | 76 | 4 | 5 | O | pos | 3; 30 | 49; 55 | 1; 7 | 3; 11 | 2; 3 | 3; 4 | WT | WT | pos | Caucasian | |
| 144 | f | 22 | 4 | 5 | A | pos | 2; 3 | 15; 35 | 3; 4 | 4; 13 | 6; 6 | 4; 4 | WT | WT | pos | Middle Eastern | |
| 145 | f | 68 | 4 | 5 | O | pos | 2; 29 | 44; 51 | 15; 16 | 11; 15 | 3; 6 | 1; 4 | | | pos | Caucasian | Hashimoto's Thyroiditis |
| 146 | m | 77 | 4 | 5 | A | pos | 1; 2 | 15; 57 | 4; 7 | 1; 7 | 3; 5 | 4; 4 | WT | WT | pos | Caucasian | |

(Continued)

**Table 1.** (Continued)

| | | | | | | | | | | | | | | | | | Comments |
|---|---|---|---|---|---|---|---|---|---|---|---|---|---|---|---|---|---|
| 147 | m | 73 | 4 | 5 | O | pos | 33; 68 | 15; 35 | 2; 4 | 13; 15 | 6; 6 | 1; 18 | WT | WT | pos | Caucasian | |
| 148 | m | 68 | 4 | 5 | A | pos | 3; 30 | 13; 14 | 6; 8 | 1; 7 | 2; 5 | 4; 4 | WT | WT | pos | Caucasian | |
| 149 | m | 63 | 4 | 4 | O | pos | 24; 26 | 51; 52 | 7; 12 | 11; 15 | 3; 6 | 4; 4 | WT | WT | pos | Caucasian | |
| 150 | m | 57 | 4 | 4 | O | pos | 2; 29 | 13; 44 | 6; 16 | 7; 7 | 2; 2 | 4; 11 | WT | WT | pos | Middle Eastern | |
| 151 | | | 5 | 5 | A | pos | 1;32 | 40;44 | 2; 2 | 1; 12 | 3; 5 | 2; 4 | WT | WT | pos | Caucasian | |
| 152 | m | 92 | 4 | 5 | A | pos | 3; 3 | 15; 47 | 3; 6 | 4; 13 | 3; 6 | 1; 5 | WT | WT | pos | Caucasian | |
| 153 | f | 66 | 4 | 4 | A | pos | 2; 68 | 18; 53 | 4; 7 | 4; 13 | 3; 6 | 4; 4 | WT | WT | pos | Middle Eastern | |
| 154 | f | 65 | 3 | 3 | AB | pos | 2; 3 | 38; 44 | 5; 12 | 11; 13 | 3; 6 | 2; 2 | het | WT | pos | Caucasian | |
| 155 | f | 71 | 4 | 4 | O | neg | 1; 68 | 44; 44 | 5; 5 | 1; 15 | 5; 6 | 4; 23 | WT | WT | pos | Caucasian | |
| 156 | m | 80 | 4 | 4 | A | neg | 3; 3 | 7; 15 | 7; 7 | 4; 4 | 3;3 | | WT | WT | pos | Caucasian | |
| 157 | m | 68 | 5 | 5 | O | neg | 11; 23 | 44; 49 | 5; 7 | 4; 13 | 2; 6 | | WT | WT | pos | Caucasian | Hypertension, carcinoma of the larynx, hypothyroidism, prostate hyperplasia, throat polyp, esophageal-pulmonary fistula |
| 158 | m | 70 | 4 | 4 | A | pos | 2;68 | 27;44 | 2;7 | 11 | 3 | 1;4 | het | WT | pos | Caucasian | |
| 159 | m | 54 | 4 | 4 | A | pos | 2;3 | 7;44 | 5;7 | 4;15 | 3;6 | 2;4 | WT | WT | pos | Caucasian | |
| 160 | f | 58 | 3 | 3 | O | pos | 2;2 | 14;44 | 5;15 | 11;15 | 3;6 | 4;23 | WT | WT | pos | Caucasian | Hypertension; allergic rhinitis |
| 161 | m | 53 | 3 | 4 | B | pos | 32;33 | 35;51 | 1;4 | 11 | 3 | 3;15 | WT | WT | pos | Caucasian | Hypertension |
| 162 | m | 29 | 3 | 4 | A | neg | | | | | | | WT | WT | pos | Caucasian | post kidney transplantation, cystinosis, hepatitis E, hypertension, allergy (atropine) |

## Genotyping for Factor V Leiden and prothrombin G20210A

The Factor V Leiden and prothrombin (20210G>A)-genotypes (in the HGVS nomenclature version 20.05 NM000130.4: c.1601G>A, p.(Arg534Gln) and NM_000506.4: c.*97G>A, respectively) were determined by sequence-specific PCR (Factor V Leiden Quicktype and Factor II 20210 G>A Quicktype, Attomol Molekulare Diagnostika GmbH, Bronkow, Germany) according to the manufacturer's instructions. The samples were analyzed in a 3% agarose gel in comparison to a blank, a negative (wildtype DNA), and a heterozygous control.

## Design of custom-made next-generation sequencing (NGS) panel

A customized NGS panel was designed to cover exonic and partially intronic regions within genes in the natural killer cell receptor family [15], the renin-angiotensin-aldosterone system, the kallikrein-kinin system, and other genes previously found to be relevant in SARS-CoV-2-infection [1]. In total, several genomic loci, spanning 144,830 bp, were targeted by sequencing. Overall, the custom-made NGS panel targeted the coding, untranslated, and splicing-regulatory intronic regions of 90 genes. For a list of the genes, we refer to Fig 2. Primer sets for the targeted regions were designed *de novo* by Illumina DesignStudio.

## NGS library preparation and sequencing (non-HLA sequencing)

The DNA library preparation was performed following the Illumina DNA Prep with Enrichment protocol (Illumina, San Diego, USA) according to the manufacturer's instructions. 600 ng of genomic input DNA was used for the generation of the libraries. The libraries were purified with AMPure XP beads (Beckman Coulter, High Wycombe, UK). Additionally, libraries and sample pools were quantified using Qubit dsDNA HS kit and Qubit 4 (Thermo Fisher Scientific, Waltham, USA), whereas the library size was analyzed with the High Sensitivity D1000 assay and 4200 Tapestation (Agilent Technologies, Palo Alto, USA). Samples were pooled reaching a final 1.4 pM library solution, which, subsequently, was loaded on the Illumina MiniSeq platform (Illumina, San Diego, USA) for (150 bp x 2) paired-end sequencing.

## Read analysis and variant calling

Reads generated from the targeted regions were aligned to the reference genome hg19/GRCh37 using the Burrows-Wheeler Aligner (0.7.7-isis-1.0.2) [16]. Indexed Sequence Alignment/Map (SAM) and its Binary Alignment/Map (BAM) file versions were computed using SAMtools (0.1.19-isis-1.0.3). Variants were called and annotated using the GATK package (v1.6-23-gf0210b3) [17]. Sequence data were processed using the GensearchNGS software suit (1.7.058, Phenosystems, Braine le Chateau, Belgium). Only samples with a mean coverage depth higher than 90 on-target passing filter reads were considered for subsequent analysis. Data were filtered based on a variant coverage of at least ten reads and a minimum variant allele frequency (VAF) of 20%. The detected variants were assessed with the dbSNP database [18] (The Genome Aggregation Database, NHLBI Exome Sequencing Project, NCBI dbSNP, Human Gene Mutation Database). Further, a variant *in silico* prediction was carried out with PolyPhen-2 [19].

## Descriptive statistics of age, gender, blood group and HLA

We wrote Python scripts (Python version 3.7.6) in Jupyter Notebook. We used modules from the scipy package (version 1.4.1) for statistical calculations and applied ML algorithms from the scikit-learn library (version 0.22.1).

| | | | | | | |
|---|---|---|---|---|---|---|
| NK receptors | Non-HLA | Co-receptors | CD59, SLAMF6, KLRF1, CD226, CD244 | | | |
| | | Inhibitory | PDCD1, SIGLEC7, CD300a, CD96, IL1RAPL1, SIGIRR, TIGIT, HAVCR2 | | | |
| | | Activating | NCR3, NCR2, NCR1, KLRK1, FCGR3A, FCGR3B | | | |
| | HLA | Inhibitory | KLRC1, KLRD1, KIR2DL1, KIR2DL3, KIRDL5, KIR3DL1, KIR3DL2, LILRB1, LAG3, KLRC3, | | | |
| | | Activating | KIR2DS1, KIR2DS2, KIR2DS3, KIRDL4, KIR2DS4, KIR2DS5, KIR3DS1, KLRC2, PVR, KLRC3, LIR | | | |
| | | Non-classical | HLA-E | | | |
| | | Homing | CCR7, CXCR3, SELL, CXCR2, CX3CR1, CMKLR1, CXCR4, CCR5, S1PR5, KIT, NCAM1, B3GAT1, CXCR1 | | | |
| Blood pressure | Renin-angiotensin-aldosterone system | | REN, AGT, ACE2, CYP11B2, ATP6AP2, HSD11B2, AGTR1, AGTR2, MAS1, RENBP, NR3C2, MTHFR, CLCN6, NPPA, NPPB, AGTRAP, TMPRSS2, PLG, KNG1, ACE | | | |
| | Kinin-kallikrein system | | MME, KNG1, KLKB1, KLK1, F12, ENPEP, BDKRB1, BDKRB2, NOS3, TAC1, CPM, SERPING1, PRCP, ACE | | | |
| Blood type | | | ABO | | | |

| Chr | Position | Variant allele | %Low_Clinic_Sc | %High_Clinic_Sc | Gene |
|---|---|---|---|---|---|
| 17 | 61554493_61554494 | insGCC | 0.00 | 1.53 | ACE |
| 17 | 615544947 | G>C | 1.07 | 0.00 | ACE |
| 17 | 61554517 | T>G | 1.07 | 0.00 | ACE |
| 17 | 61554520 | T>G | 1.07 | 1.53 | ACE |
| 17 | 61554556 | T>G | 1.08 | 0.00 | ACE |
| 17 | 61554559 | A>C | 2.10 | 0.00 | ACE |
| 17 | 61554681 | A>C | 0.00 | 1.50 | ACE |
| 17 | 61554696 | A>C | 2.15 | 1.53 | ACE |
| 17 | 61554703 | A>T | 1.07 | 0.00 | ACE |
| 17 | 61554706 | T>G | 9.67 | 4.61 | ACE |
| 17 | 61554721 | G>C | 1.07 | 0.00 | ACE |
| 17 | 61554724 | C>G | 0.00 | 3.07 | ACE |
| 17 | 61562440 | A>T | 0.00 | 7.69 | ACE ACE |
| 17 | 61562445 | A>T | 1.07 | 13.84 | ACE ACE |
| 17 | 61568575 | G>A | 1.07 | 0.00 | ACE ACE |
| 19 | 51650050 | T>G | 0.00 | 1.53 | SIGLEC7 |
| 1 | 169672589 | G>T | 40.80 | 46.15 | SELL C1orf112 |
| 1 | 169672595 | G>T | 9.67 | 7.69 | SELL C1orf112 |
| 1 | 169673836 | A>G | 1.07 | 0.00 | SELL C1orf112 |
| 1 | 169679563_169679567 | delATATC | 1.07 | 0.00 | SELL C1orf112 |
| 21 | 42870031 | T>C | 0.00 | 1.53 | TMPRSS2 |
| 21 | 42879896 | A>C | 0.00 | 1.53 | TMPRSS2 |
| 21 | 42879898 | A>C | 0.00 | 4.61 | TMPRSS2 |
| 21 | 42879914 | G>A | 0.00 | 1.53 | TMPRSS2 |
| 21 | 42879923 | A>T | 1.07 | 0.00 | TMPRSS2 |
| 21 | 42879931 | T>G | 1.07 | 0.00 | TMPRSS2 |
| 21 | 42879932 | C>G | 1.07 | 0.00 | TMPRSS2 |
| 21 | 42879935 | C>G | 1.07 | 0.00 | TMPRSS2 |
| 21 | 42879943 | A>C | 1.07 | 1.50 | TMPRSS2 |
| 21 | 42879951 | A>T | 1.07 | 0.00 | TMPRSS2 |
| 3 | 45834213 | A>C | 1.07 | 0.00 | SLC6A20 |
| 3 | 45834994 | C>A | 7.50 | 4.61 | SLC6A20 |

**Fig 2. Genes targeted in the next-generation sequencing panel and list of rare variants.**

## Association testing

The focus of our analyses was on the genes *SIGLEC7*, *ACE*, *SELL*, *TMPRSS2*, and *SLC6A20*, since they were found to be associated with severity of SARS-CoV-2 infection in recent studies [1, 6, 7]. To explore SNPs, which lead to increased severity and to find novel SNPs, we conducted an association analysis.

SNPs that occurred in less than four patients were filtered out and excluded by the analysis. Testing significance, the p-values for the SNPs were calculated using logistic regression and adjusted for the covariates gender and age. The open-source whole-genome association analysis toolset PLINK (version 1.9) [19] was used to build the regression model. To adjust for multiple testing, the p-values were corrected using the false discovery rate approach by Benjamini-Hochberg [20].

To consider associations between all the variants within each gene, gene-based tests were applied. Accounting for multiple independent functional variants may increase the power to identify disease-associated genes. By aggregating the information on all the variants, we obtained a single p-value that corresponded to the significance of the association of the gene. We applied the versatile gene-based test for genome-wide association studies (VEGAS) [21] and the gene-based association test with extended Simes procedure (GATES) [22]. Concerning the role of HLA genotypes, the dataset limited power for the calculation of significant associations, similar to the statistical analysis by Schetelig *et al.* [23].

## Analysis of mutation impact on protein structure

To find the SNP-containing structures in the Protein Data Bank (PDB) [24], we used UniProt [25]. We applied the SWISS-MODEL repository and the related analysis and visualization tool [26] to check the structural coverage. For unknown structures and regions without structural coverage, we analyzed predicted structures based on the AlphaFold database [27, 28]. For each predicted structure, AlphaFold provides a color-coded confidence score per residue to evaluate the prediction quality. To estimate the impact of a SNP on protein structure and function we ran the PolyPhen-2 software [19].

PolyPhen-2 annotates coding and nonsynonymous SNPs, and it predicts damaging missense mutations, using sequence-based and structure-based predictive features. To train and test PolyPhen-2, two pairs of datasets were generated. The first pair, HumDiv, is based on all 3,155 damaging alleles annotated in UniProt as causing human Mendelian diseases and affecting protein stability or function, as well as on 6,321 differences between human proteins and their closely related mammalian homologs, which have been assumed to be non-damaging. The second pair, HumVar3, consists of all the 13,032 human disease-causing mutations from UniProt and 8,946 human nonsynonymous single-nucleotide polymorphisms (nsSNPs) without annotated involvement in disease, which have been treated as non-damaging. PolyPhen-2 qualitatively classifies a mutation as benign, possibly damaging, or probably damaging, scoring with a value between 0 and 1, with 1 for damaging impact and 0 for non-damaging impact of the SNP on protein structure.

## Integration of other studies

32 SNPs of five genes of interest, *ACE*, *SIGLEC7*, *SELL*, *TMPRSS2*, and *SLC6A20*, were rare variants and not reported in other datasets such as gnomAD (version 2.1.1) [29] or dbSNP. For a list of the 32 rare variants, we refer to Fig 2.

## Results

### Patient cohort

We gathered a dataset of patients with SARS-CoV-2 infection. For an outline of the experimental workflow and sequencing, we refer to Fig 1. Fig 2 lists 32 SNPs of the five genes *ACE*, *SIGLEC7*, *SELL*, *TMPRSS2*, and *SLC6A20* that are rare variants and which had not been reported previously in other datasets such as gnomAD. Fig 3 depicts distributions of age,

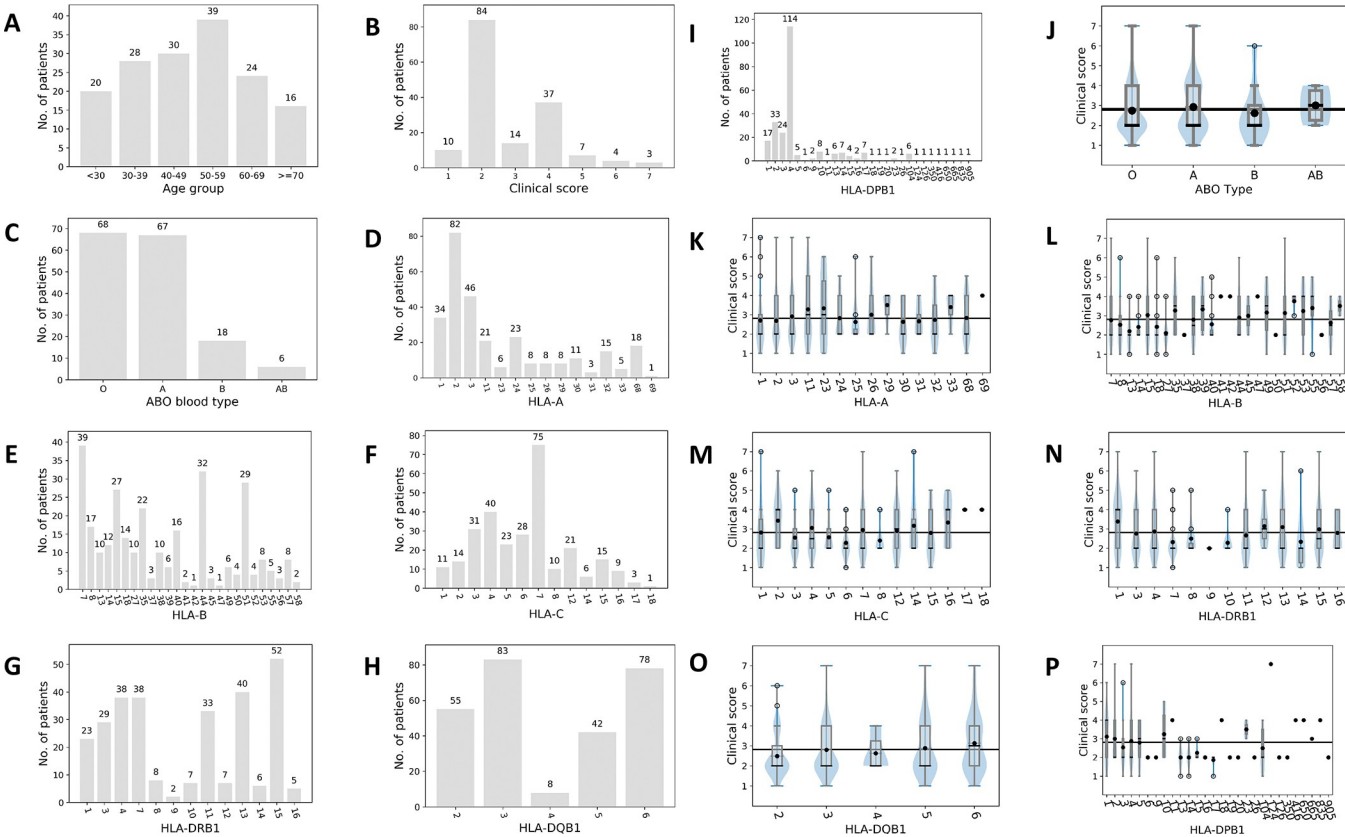

**Fig 3. Descriptive statistics.** (A) The number of patients within age groups. (B) The number of patients versus clinical score. (C) The number of patients versus ABO blood type. (D) The number of patients versus HLA-A type. (E) The number of patients versus HLA-B type. (F) The number of patients versus HLA-C type. (G) The number of patients versus HLA-DRB1 type. (H) The number of patients versus HLA-DQB1 type. (I) The number of patients versus HLA-DPB1 type. (J) Mean clinical score versus ABO blood type. (K) Mean clinical score versus HLA-A type. (L) Mean clinical score versus HLA-B type. (M) Mean clinical score versus HLA-C type. (N) Mean clinical score versus HLA-DRB1 type. (O) Mean clinical score versus HLA-DQB1 type. (P) Mean clinical score versus HLA-DPB1 type. (J-O) The black line shows the overall mean of the clinical score of 2.81. The black bullets show the mean clinical score for patient groups. Violin plots superimposed by box plots indicate the densities of clinical scores. We found no statistical significance for any of the subgroups in J-P regarding the deviation of their mean clinical score from the overall mean clinical score of 2.81, i.e. Mann-Whitney U tests gave significance values larger than 5%.

clinical score, ABO blood type, and HLA type for the patient cohort. We found no statistical significance for clinical scores of patients with any subgroup of ABO or HLA type; i.e., Mann-Whitney U tests led to significance levels greater than 5%.

Fig 4 illustrates the role of age and gender in the clinical score. In the jitter plot in Fig 4A the red and blue dots represent data points for female and male patients, respectively. The grey curve (unisex) shows the increase in mean clinical score for all patients. The mean clinical score increases with age. A high clinical score is associated with high age (Kendall tau 0.41, p = 1.2 $10^{-12}$, Stuart-Kendall Tau-c test). The unisex curve has its maximal slope of 0.1874 +/- 0.0495 [clinical score/year] at age 60.394 +/- 3.466 years. The blue curve (male) shows the increase in mean clinical score for male patients. The blue curve has its maximal slope of 0.245 +/- 0.124 [clinical score/year] at age 55.9 +/- 2.8 years. A high clinical score in male patients is associated with higher age (Kendall tau 0.41, p = 1.2 $10^{-7}$, Stuart-Kendall Tau-c test). The red curve (female) shows the increase in mean clinical score for female patients. The curve for females is steeper than the curve for males and has its maximal slope of 0.942 +/- 0.552 [clinical score/year] at age 66.1 +/- 1.0 years. A high clinical score of female patients is associated with

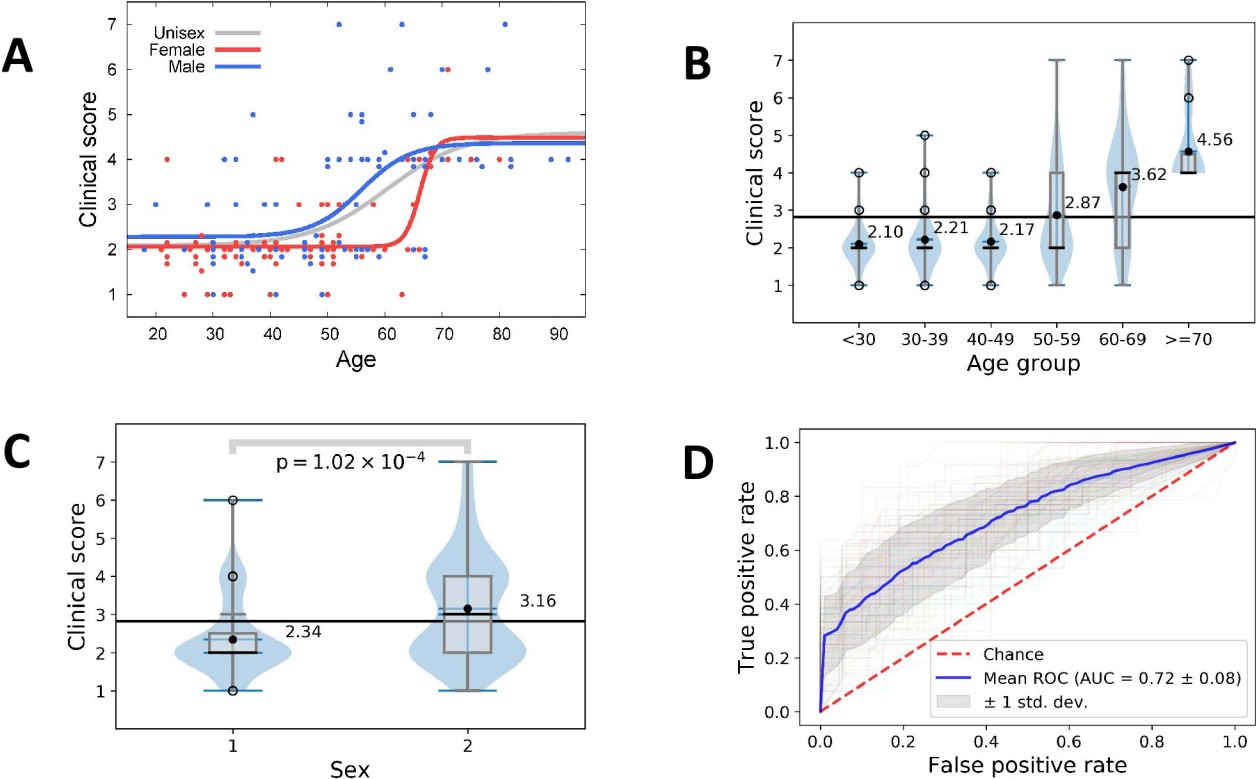

**Fig 4. Significance of age and gender.** (A) Jitter plot of clinical score versus age. (B) Mean clinical score versus age group. (C) Mean clinical scores of female versus male patients of all ages. (B, C) The black line shows the overall mean of the clinical score of 2.81. The black bullets show the mean clinical score for patient groups. Violin plots superimposed by box plots indicate the densities of clinical scores. (D) Receiver operating characteristics (ROC) of a random forest prediction of hospitalization, i.e., clinical score 4–7, based on age and gender.

higher age (Kendall tau 0.34, p = 1.0 $10^{-4}$, Stuart-Kendall Tau-c test). For young (age < 40) and old (age > 70) individuals, the clinical scores of female and male patients were not significantly different (Mann–Whitney U test). For age > 50 years, the mean clinical score increases monotonically with age (Fig 4B), and the clinical scores differ significantly from younger patients (age < 50, Mann–Whitney U test). The clinical scores of females are significantly lower than those of males (Mann-Whitney U test, p-value 1.02E-4) (Fig 4C). Fig 4D shows the receiver operating characteristics (ROC) of a random forest prediction of hospitalization, i.e., clinical score 4–7, based on age and gender. The blue line is the mean ROC curve of a Monte Carlo cross-validation with 100 random splits into 70% training and 30% test sets. The shaded area, light gray, indicates the standard deviation. The mean area under the curve (AUC) value of 0.72± 0.08 demonstrates the power of age and gender in predicting the hospitalization of a patient. For a discussion of the role of gender and age on Covid-19 severity, we refer to [30].

## Panel sequencing quality

On average more than 1.2 M of passing-filter (PF) reads were generated per sample with 89.1% of them in Q3. Similarly, more than 1.2 M reads aligned to the reference genome representing 98.7% of all PF reads, generating a mean coverage of 332.6x per sample with a coverage uniformity of 98.7%.

Sample sequence-containing and base quality score files (FastQ) were aligned to the reference genome and transformed into SAM files and their binary counterpart, BAM files. The

latter were subjected to variant calling and annotation, generating vcf files, calling an average of 199.69 single-nucleotide variants (SNVs) per sample with 89.75% of these being annotated in dbSNP. Fig 5A shows boxplots of the numbers of SNPs per patient, Fig 5B shows numbers of hemizygous, heterozygous, and homozygous SNPs per patient, and Fig 5C shows numbers of SNPs for patients with a low versus high clinical score.

## Rare variants and structural properties

SNPs are associated with disease severity in infection with SARS-CoV-2. An average of approximately 220 SNPs per patient was collected and analyzed (Fig 5A). The majority of variants were heterozygous or homozygous, and only a few were hemizygous (Fig 5B). A slight enrichment of missense variants was observed in patients with high clinical score (Fig 5C). Specifically, a mean of 81.3 missense SNPs was determined for patients with low clinical score versus 83.1 missense SNPs for patients with high clinical score, whereby the distribution in the latter group was more broad.

The statistical significance of the association between variants and disease severity were calculated by PLINK logistic regression, and the p-values were corrected for the covariates age and gender. Four SNPs characterized by the strongest association with clinical score are listed in Table 2. The SNPs are located in the genes *ACE*, *SIGLEC7*, *TMPRSS2*, and *SLC6A20*. Three of them (rs993496436, rs3787950, rs2276858) are previously known, but the variant of the gene *ACE*, which is located on chromosome 17 at position 61562445, is not present in the dbSNP database.

10 patients, of whom 70% were male and 30% were female, were identified with a SNP in the *ACE* gene. 80% of these patients had a clinical score 4, and 40% of these patients were in the oldest age group (60–69 years). 59 patients had a SNP in the *SIGLEC7* gene with a fairly even distribution between men (55%) and women (45%). 66% of these patients had a low clinical score of 2, and the distribution amongst the age groups was almost even. A SNP in the *TMPRSS2* gene was identified in 30 patients (67% men, 33% women). 46% of patients had a clinical score 2, whereby the other clinical scores, as well as the representation of age groups was fairly well distributed. 12 patients (73% men, 27% female) were found to have a SNP in the *SLC6A20* gene. The majority of patients (67%) were found to have a clinical score of 2, and 46% of patients with a SNP in this gene were in the age group 40–49 years.

The four SARS-CoV-2-related genes may be risk variants, which individually, however, were not strongly associated with disease severity. The size of our data set (159 patients) was too small to reach a false discovery rate below 5%. To increase the power of the analysis and to identify the aggregated risk of variants on a gene level, when simultaneously considering the whole set of SNPs, we applied gene-based tests. The cumulative association of genes *SIGLEC7*, *ACE*, *TMPRSS2*, *SELL*, and *SLC6A20* are listed in Table 3. The number of SNPs per gene varies and is provided in the second column, "SNPs per gene". False discovery rates (with correction for multiple testing), which were determined by GATES and VEGAS tests, are shown. Genes *SIGLEC7* and *ACE* appear to have significant false discovery rates (p-value < 0.05). The significance for gene *SIGLEC7* is associated with a group of patients who presented with at least one of two SNPs. The significance for gene *ACE* is associated with a group of patients who presented with at least one of twelve SNPs.

Given the significant association of the *SIGLEC7* gene with clinical score (Table 3), we explored the protein structure and function of the coding nonsynonymous SNP rs993496436 of this gene, hypothesizing that the SNP rs993496436 might be relevant for disease severity in infection with SARS-CoV-2. The relevance of SNP rs993496436 had been previously suggested by Sharif-Askari et al. [6].

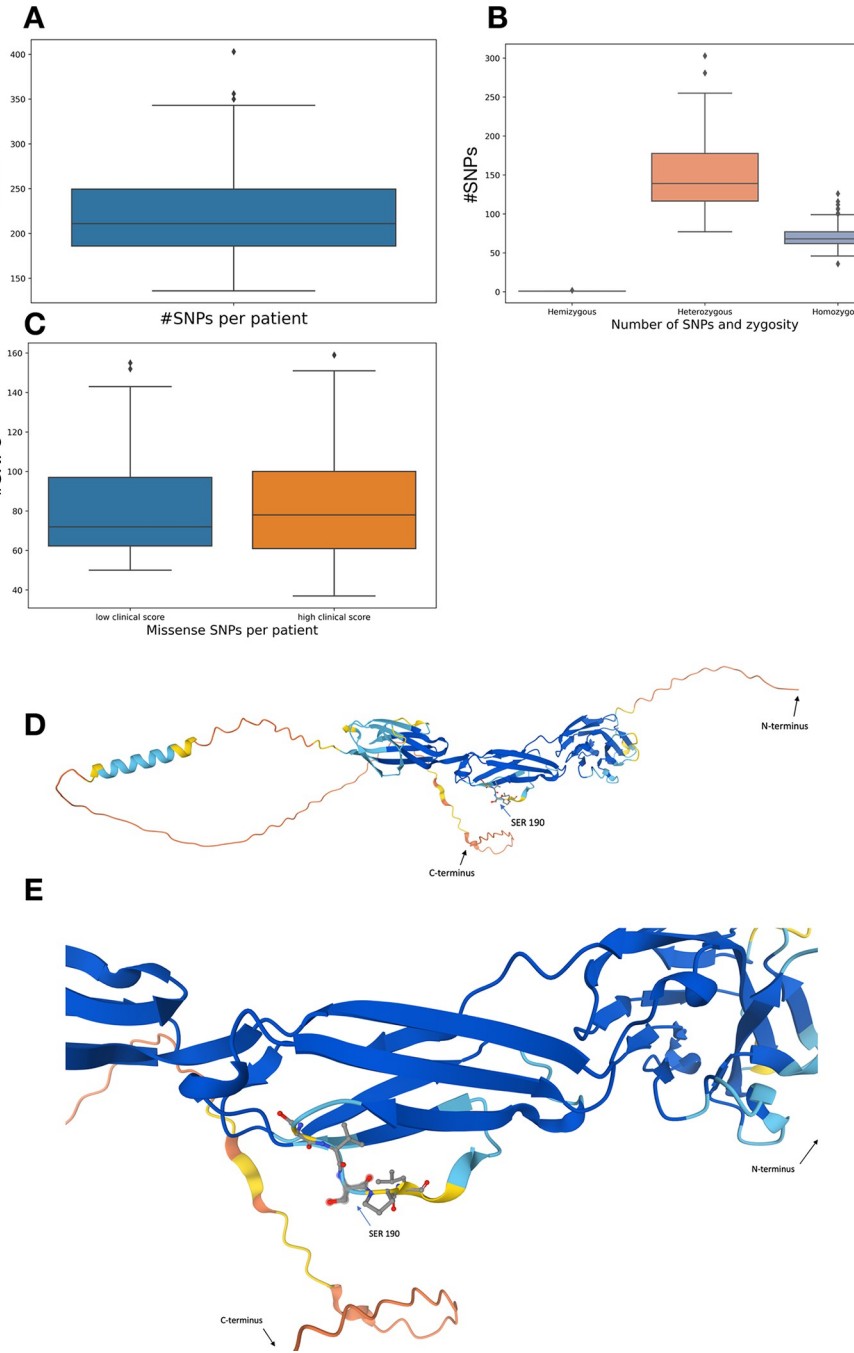

**Fig 5. Single-nucleotide polymorphisms in patient cohort.** A) Number of SNPs per patient. An average of approximately 220 SNPs per patient was detected. B) Comparison of the number of SNPs with zygosity. C) Comparison of the occurrence of missense SNPs in the low versus high clinical score. D) The whole AlphaFold-predicted structure, AF-Q9Y286-F1, of the nonsynonymous SNP, rs993496436, encoding the protein sialic acid-binding Ig-like lectin 7 (SIGLEC7) in humans (Q9Y286). The mutated serine is illustrated in a stick-and-ball representation. It is located within the light blue-colored area in ribbon representation, exhibiting a confidence score of 70 to 90, whereby 100 is the best score. The figure was generated using AlphaFold software. E) Magnification of the mutated serine illustrated in a stick-and-ball representation as in D). The mutated serine is located within a light blue-colored range, standing for a confidence score of 70 to 90.

**Table 2. List of SNPs with the strongest association with clinical score.** The p-values are not corrected for multiple testing. The size of our data set (159 patients) is too small to reach a false discovery rate below 5%.

| Chr | Start | Variant allele | dbSNP ID | Gene | Occurrence mild/severe | p-value |
|-----|-------|----------------|----------|------|------------------------|---------|
| 17 | 61562445 | A>T | no | *ACE* | 1/9 | 0,002 |
| 19 | 51647798 | C>G | rs993496436 | *SIGLEC 7* | 45/14 | 0,005 |
| 21 | 42866296 | T>C | rs3787950 | *TMPRSS2* | 15/15 | 0,039 |
| 3 | 45813993 | G>A | rs2276858 | *SLC6A20* | 10/2 | 0,045 |

The SNP in the *SIGLEC7* gene represents a mutation from cytosine to guanine. During transcription to a chain of amino acids, this SNP causes a change from serine to cysteine at position 190. Based on the corresponding UniProt entry, Q9Y286, we found six protein structures in the PDB database (1NKO, 1O7S, 1O7V, 2DF3, 2G5R, 2HRL) but none of them covered position 190. AlphaFold predicted a structure, AF-Q9Y286-F1, which covered the sequence from 1 to 467 and hence, included position 190 of the SNP (Fig 5D and 5E).

AlphaFold computed a middle to low confidence score for prediction of the structure in the region around position 190. The mutation is located in a loop region of a beta-sheet. The flexibility of loop regions makes it difficult to estimate how the mutation affects the structure and function of SIGLEC7. The software PolyPhen-2 predicted the mutation to be benign for both training sets HumDiv and HumVar with a score of 0.447 and 0.309, respectively (for the PolyPhen-2 report, S1 Fig). The mutation causes a change of residues from serine to cysteine. Serine and cysteine exhibit very different chemical features, so that–given the location of the amino acid change at the surface of the protein in a flexible loop region–one may speculate that the mutation may induce a significant change in structure and function of the protein.

## Discussion

Since the beginning of the pandemic, several manuscripts have been published on risk factors associated with a mild versus a severe course of infection with SARS-CoV-2 [31–35]. Our limited sample size did not allow any definitive conclusions on genetic risk factors such as ABO or HLA type directly influencing clinical course. Regardless, though small in nature, our study has value in providing a dataset on patient demographics, blood, and HLA type, as well as SNPs in genes of the NK cell receptor, renin-angiotensin-aldosterone and kallikrein-kinin systems in correlation with their respective clinical course. In addition, we provide a case study and protein structure analysis on a SNP in the *SIGLEC7* gene, which was significantly associated with clinical score.

Our finding of no significant association of HLA-A, -B, -C, and -DRB1 genotypes with the severity of infection with SARS-CoV-2 was in accordance with the results of another study [23]. Also for HLA-DQB1, and -DPB1 genotypes, which were not studied by Schetelig et al.

**Table 3. Gene-based association test for the five SARS-CoV-2-related genes *SIGLEC7, ACE, TMPRSS2, SELL,* and *SLC6A20*.** The number of SNPs per gene considered for the test varies. False discovery rates (with correction for multiple testing) computed by two tests, GATES and VEGAS, are provided.

| Gene | SNPs per gene | GATES | VEGAS |
|------|---------------|-------|-------|
| *SIGLEC7* | 2 | 0,011 | 0,012 |
| *ACE* | 13 | 0,021 | 0,020 |
| *TMPRSS2* | 8 | 0,220 | 0,254 |
| *SELL* | 6 | 0,483 | 0,613 |
| *SLC6A20* | 16 | 0,534 | 0,398 |

(2021), we identified no genotype as a major risk factor. As the study by Schetelig et al. (2021) corrected for gender, both age and age squared, both body mass index (BMI) and BMI squared, the intake of medication for diabetes mellitus or arterial hypertension and smoking status, a direct comparison of the results of our study and the study by Schetelig et al. (2021) is not possible. Both studies may suffer from insufficient prediction power to determine the role of HLA genotypes, as a much larger data set would be required to reach conclusive statements for the role of HLA genotypes. In contrast, several studies, including ours, discovered an association between male gender and age with more severe clinical course [30, 36, 37].

SIGLEC 7, which acts as an inhibitor of the cytotoxicity of NK cells and was found to be associated with clinical course in our study, may be involved in the clinical course of infection with SARS-CoV-2 by its ability to mediate sialic-dependent binding to cells or to block signal transduction via dephosphorylation of signaling molecules [38]. Given the position of the SNP in the gene and its corresponding amino acid alteration, the mutation may affect the clinical course via structural and, consequently, functional changes in the protein. In addition, ACE2, a homologue of ACE, is the recognized receptor for SARS-CoV-2 [39], and infection of cells by SARS-CoV-2 is believed to alter ACE/ACE2 balance [40]. Our association of SNPs in the *ACE* gene with a clinical score of 4 and this SNP's higher incidence in males of the oldest age group in our study may confirm the pathophysiologic role of ACE/ACE2 in infection with SARS-CoV-2. However, these data will need to be further validated mechanistically in future studies.

In conclusion, our work provides SNP data in immunoregulatory genes in patients with SARS-CoV-2 infection, stratified by clinical course. We present a structural analysis on the SIGLEC7 protein, in which a SNP we found to impact clinical course, is expected to alter protein structure and function. It is to be hoped that our findings may encourage further work on the largely still obscure functions of SIGLEC7 and help predict clinical outcomes in patients with SARS-CoV-2 infection and possibly other diseases.

## Supporting information

**S1 Fig. PolyPhen-2 report for SNP rs993496436 in the *SIGLEC7* gene.** The prediction of SNP rs993496436 is benign. The SNP causes a residue change from serine to cysteine at position 190 in the amino acid chain.
(PDF)

**S1 Table. For each participant, S1 Table lists the SNPs and zygosity information (0: homozygous for the reference, 1: heterozygous, 2: homozygous for the alternate).** S1 Table is also available online in the open data repository figshare, https://doi.org/10.6084/m9.figshare.20068868.v2. The title of the file is "*SNPs per patient and zygosity information*".
(XLSX)

## Author Contributions

**Conceptualization:** Nikoletta Katsaouni, Ina Koch, Marcel H. Schulz, Daniela S. Krause.

**Data curation:** Pablo Llavona, Yascha Khodamoradi, Stephanie Körber, Christof Geisen, Christian Seidl, Maria J. G. T. Vehreschild, Sandra Ciesek, Jörg Ackermann, Daniela S. Krause.

**Formal analysis:** Nikoletta Katsaouni, Pablo Llavona, Jörg Ackermann, Marcel H. Schulz, Daniela S. Krause.

**Funding acquisition:** Ina Koch, Marcel H. Schulz, Daniela S. Krause.

**Investigation:** Jörg Ackermann, Marcel H. Schulz.

**Methodology:** Nikoletta Katsaouni, Pablo Llavona, Jörg Ackermann, Ina Koch, Marcel H. Schulz, Daniela S. Krause.

**Project administration:** Ina Koch, Marcel H. Schulz, Daniela S. Krause.

**Software:** Nikoletta Katsaouni, Ann-Kathrin Otto.

**Supervision:** Ina Koch, Marcel H. Schulz, Daniela S. Krause.

**Validation:** Nikoletta Katsaouni, Ina Koch, Marcel H. Schulz, Daniela S. Krause.

**Visualization:** Nikoletta Katsaouni, Ann-Kathrin Otto, Jörg Ackermann, Ina Koch.

**Writing – original draft:** Nikoletta Katsaouni, Jörg Ackermann, Ina Koch, Marcel H. Schulz, Daniela S. Krause.

**Writing – review & editing:** Nikoletta Katsaouni, Jörg Ackermann, Ina Koch, Marcel H. Schulz, Daniela S. Krause.

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
