## [Decision Letter · Decision Letter 0]

8 Mar 2023

PONE-D-23-03131Dataset of single nucleotide polymorphisms of immune-associated genes in patients with SARS-CoV2 infectionPLOS ONE

Dear Dr. Katsaouni,

Thank you for submitting your manuscript to PLOS ONE. After careful consideration, we feel that it has merit but does not fully meet PLOS ONE’s publication criteria as it currently stands. Therefore, we invite you to submit a revised version of the manuscript that addresses the points raised during the review process. Your manuscript has been reviewed and requires modifications prior to making a decision. The comments of the reviewers are included at the bottom of this letter. Reviewers indicated that methods and results sections should be improved. We would be glad to consider a substantial revision of your work, where the reviewers' comments will be carefully addressed one by one.

We look forward to receiving your revised manuscript.

Kind regards,

Asli Suner Karakulah, PhD

Academic Editor

PLOS ONE

Journal Requirements:

"This work was supported by the Goethe-Corona-Funds of the Goethe University Frankfurt to D.S.K. We acknowledge funding from the Alfons und Gertrud Kassel-Stiftung as part of the center for data science and AI and the DFG Cluster of Excellence Cardio Pulmonary Institute (CPI) [EXC 2026]. We also acknowledge funding from the consortia ACLF-I (Acute Liver Failure - Initiative) and ENABLE (Unraveling mechanisms driving cellular homeostasis, inflammation and infection to enable new approaches in translational medicine) (Hessian Ministry of the Arts and Sciences)."

"This work was supported by the Goethe-Corona-Funds of the Goethe University Frankfurt to D.S.K. We acknowledge funding from the Alfons und Gertrud Kassel-Stiftung as part of the center for data science and AI and the DFG Cluster of Excellence Cardio Pulmonary Institute (CPI) [EXC 2026]. We also acknowledge funding from the consortia ACLF-I (Acute Liver Failure - Initiative) and ENABLE (Unraveling mechanisms driving cellular homeostasis, inflammation and infection to enable new approaches in translational medicine) (Hessian Ministry of the Arts and Sciences)."

We note that one or more of the authors is affiliated with the funding organization, indicating the funder may have had some role in the design, data collection, analysis or preparation of your manuscript for publication; Goethe University Frankfurt 

In other words, the funder played an indirect role through the participation of the co-authors. If the funding organization did not play a role in the study design, data collection and analysis, decision to publish, or preparation of the manuscript and only provided financial support in the form of authors' salaries and/or research materials, please do the following:

(1) Review your statements relating to the author contributions, and ensure you have specifically and accurately indicated the role(s) that these authors had in your study. These amendments should be made in the online form.

(2) Confirm in your cover letter that you agree with the following statement, and we will change the online submission form on your behalf: 

“The funder provided support in the form of salaries for authors [insert relevant initials], but did not have any additional role in the study design, data collection and analysis, decision to publish, or preparation of the manuscript. The specific roles of these authors are articulated in the ‘author contributions’ section.”"

"This work was supported by the Goethe-Corona-Funds of the Goethe University Frankfurt to

D.S.K. We acknowledge funding from the Alfons und Gertrud Kassel-Stiftung as part of the

center for data science and AI and the DFG Cluster of Excellence Cardio Pulmonary Institute

(CPI) [EXC 2026]. We also acknowledge funding from the consortia ACLF-I (Acute Liver Failure

- Initiative) and ENABLE (Unraveling mechanisms driving cellular homeostasis, inflammation

and infection to enable new approaches in translational medicine) (Hessian Ministry of the

Arts and Sciences)."

"This work was supported by the Goethe-Corona-Funds of the Goethe University Frankfurt to D.S.K. We acknowledge funding from the Alfons und Gertrud Kassel-Stiftung as part of the center for data science and AI and the DFG Cluster of Excellence Cardio Pulmonary Institute (CPI) [EXC 2026]. We also acknowledge funding from the consortia ACLF-I (Acute Liver Failure - Initiative) and ENABLE (Unraveling mechanisms driving cellular homeostasis, inflammation and infection to enable new approaches in translational medicine) (Hessian Ministry of the Arts and Sciences)."

6. Thank you for stating the following in your Competing Interests section: "NO"

7. In your Data Availability statement, you have not specified where the minimal data set underlying the results described in your manuscript can be found. PLOS defines a study's minimal data set as the underlying data used to reach the conclusions drawn in the manuscript and any additional data required to replicate the reported study findings in their entirety. All PLOS journals require that the minimal data set be made fully available. For more information about our data policy, please see http://journals.plos.org/plosone/s/data-availability.

8. We note that you have stated that you will provide repository information for your data at acceptance. Should your manuscript be accepted for publication, we will hold it until you provide the relevant accession numbers or DOIs necessary to access your data. If you wish to make changes to your Data Availability statement, please describe these changes in your cover letter and we will update your Data Availability statement to reflect the information you provide.

9. Please include a new copy of Table 1 in your manuscript; the current table is difficult to read. Please follow the link for more information: https://blogs.plos.org/plos/2019/06/looking-good-tips-for-creating-your-plos-figures-graphics/

Reviewers' comments:

Reviewer's Responses to Questions

**Comments to the Author**

1. Is the manuscript technically sound, and do the data support the conclusions?

Reviewer #1: Yes

Reviewer #2: Yes

Reviewer #3: Yes

2. Has the statistical analysis been performed appropriately and rigorously? 

Reviewer #1: Yes

Reviewer #2: Yes

Reviewer #3: Yes

3. Have the authors made all data underlying the findings in their manuscript fully available?

Reviewer #1: Yes

Reviewer #2: Yes

Reviewer #3: Yes

4. Is the manuscript presented in an intelligible fashion and written in standard English?

Reviewer #1: Yes

Reviewer #2: Yes

Reviewer #3: No

5. Review Comments to the Author

Reviewer #1: In this paper, the authors are proposed “Dataset of single nucleotide polymorphisms of immune-associated genes in patients with SARS-CoV2 infection”

The strengths of the paper are that it is well structured, the description of the related work is well done and that results are extensively compared to results of the similar research.

Minor revisions:

1. Authors should draw a graphical abstract of the proposed approach

2. Authors should justify the proposed approach.

3. Proofread the entire manuscript.

4. Authors should submit dataset sample in supplementary files, and some supplementary files are not open.

Reviewer #2: Dear authors,

First of all, I congratulate you for doing this fascinating study.

The article itself is well written, although it needs some corrections, as I mentioned in manuscript using track changes.

Please use the italic format for the name of genes and normal font for the protein’s names.

Success in your further research

Reviewer #3: It is necessary that the authors mention in the methods section, the program with which they analyzed the data of age, sex, blood group and HLA.

Likewise, to enrich the results, it is advisable to mention the clinical characteristics of the patients, mainly those who presented the 4 SNPs, as well as discuss the clinical data to reinforce the importance of the study.

It is necessary to unify universal terminology, such as SARS-CoV-2, COVID-19, among others throughout the article.

6. PLOS authors have the option to publish the peer review history of their article (what does this mean?). If published, this will include your full peer review and any attached files.

Reviewer #1: No

Reviewer #2: **Yes: **Mohadeseh Haji Abdolvahab

Reviewer #3: **Yes: **Gustavo J. Vazquez-Zapien

---

## [Author Response · Author response to Decision Letter 0]

26 May 2023

We would like to thank the reviewers for reading the manuscript, for the kind response and for their helpful suggestions. In the revised version, changes are highlighted in blue boldface letters.

Reviewer #1: 

In this paper, the authors are proposed “Dataset of single nucleotide polymorphisms of immune-associated genes in patients with SARS-CoV2 infection”. The strengths of the paper are that it is well structured, the description of the related work is well done and that results are extensively compared to results of the similar research.

 Response: We thank the reviewer for these positive comments. 

Minor revisions:

Authors should draw a graphical abstract of the proposed approach

 Response: In the original manuscript we had provided a figure on the workflow of our study. However, in response to this reviewer’s suggestion we now also include an updated graphical abstract.

Authors should justify the proposed approach.

 Response: We agree with the reviewer and have now added sentences to this effect to the introduction (page 5).

Proofread the entire manuscript.

 Response: We agree with the reviewer and apologize for any previous typos. After making the requested changes, the manuscript has now been proofread again.

Authors should submit dataset sample in supplementary files, and some supplementary files are not open.

 Response: We apologize for this. We now include the entire dataset in supplementary files and have checked that they can be opened. 

Reviewer #2: 

Dear authors, First of all, I congratulate you for doing this fascinating study. The article itself is well written, although it needs some corrections, as I mentioned in manuscript using track changes.

 Response: We thank the reviewer for these encouraging words. We also thank you for explicitly pointing out errors and misleading sentences in the manuscript. Your work was a great help in the revision of the manuscript. We carefully proofread the final version. We made changes in the text according to your suggestions marked in the manuscript. Please see the list of changes below:

Page 15/16: The following sentence easily can be misunderstood:

"SIGLEC7 and ACE appear to have significant false discovery rates (p-value < 0.05 ) with 2 and 12 considered SNPs, respectively."

We changed the text to:

"Genes SIGLEC7 and ACE appear to have significant false discovery rates (p-value < 0.05). The significance for the gene SIGLEC7 is associated with a group of patients who presented with at least one of two SNPs. The significance for gene ACE is associated with a group of patients who presented with at least one of twelve SNPs."

 Page 16: The following sentence easily can be misunderstood:

"The SNP in the SIGLEC7 gene exhibited a mutation from cytosine to guanine, causing a residue change from serine to cysteine at position 190 in the amino acid chain."

We changed the text to:

"The SNP in the SIGLEC7 gene represents a mutation from cytosine to guanine. During transcription to a chain of amino acids, this SNP causes a change from serine to cysteine at position 190."

Please use the italic format for the name of genes and normal font for the protein’s names.

 Response: We agree with the reviewer and apologize. All gene names have been italicized in capital letters, while protein names are written in capital letters and are not italicized.

Success in your further research

 Thank you!

Reviewer #3: 

It is necessary that the authors mention in the methods section, the program with which they analyzed the data of age, sex, blood group and HLA.

 Response: We agree with the reviewer and thank him/her for raising this point. On page 10 we have now added the following text:

"Descriptive statistics of age, gender, blood group, and HLA

For the analysis of statistics we wrote Python scripts (Python version 3.7.6) in Jupyter Notebook. We used modules from the scipy package (version 1.4.1) for statistical calculations and applied ML algorithms from the scikit-learn library (version 0.22.1)." 

Likewise, to enrich the results, it is advisable to mention the clinical characteristics of the patients, mainly those who presented the 4 SNPs, as well as discuss the clinical data to reinforce the importance of the study.

Response: We agree with the reviewer and thank him/her for this excellent suggestion. We have added details to the results section on page 15/16 and have discussed these data on page 19. We also politely refer the reader to our table with the patients’ metadata.

It is necessary to unify universal terminology, such as SARS-CoV-2, COVID-19, among others throughout the article.

Response: We agree with the reviewer. We have unified language with respect to SARS-CoV2 and other terms, as suggested by this reviewer.

---

## [Decision Letter · Decision Letter 1]

12 Jun 2023

Dataset of single nucleotide polymorphisms of immune-associated genes in patients with SARS-CoV2 infection

PONE-D-23-03131R1

Dear Dr. Katsaouni,

We’re pleased to inform you that your manuscript has been judged scientifically suitable for publication and will be formally accepted for publication once it meets all outstanding technical requirements.

Kind regards,

Asli Suner Karakulah, PhD

Academic Editor

PLOS ONE

Additional Editor Comments (optional):

The authors addressed the reviewers' concerns and substantially improved the content of MS.

So, based on my own assessment as an academic editor, MS can be accepted in its current form.

Reviewers' comments:

Reviewer's Responses to Questions

**Comments to the Author**

1. If the authors have adequately addressed your comments raised in a previous round of review and you feel that this manuscript is now acceptable for publication, you may indicate that here to bypass the “Comments to the Author” section, enter your conflict of interest statement in the “Confidential to Editor” section, and submit your "Accept" recommendation.

Reviewer #2: All comments have been addressed

Reviewer #3: All comments have been addressed

2. Is the manuscript technically sound, and do the data support the conclusions?

Reviewer #2: Yes

Reviewer #3: Yes

3. Has the statistical analysis been performed appropriately and rigorously? 

Reviewer #2: Yes

Reviewer #3: Yes

4. Have the authors made all data underlying the findings in their manuscript fully available?

Reviewer #2: Yes

Reviewer #3: Yes

5. Is the manuscript presented in an intelligible fashion and written in standard English?

Reviewer #2: Yes

Reviewer #3: Yes

6. Review Comments to the Author

Reviewer #2: (No Response)

Reviewer #3: Dear Author.

Previous comments or suggestions have been answered and added to the manuscript. Thank you.

7. PLOS authors have the option to publish the peer review history of their article (what does this mean?). If published, this will include your full peer review and any attached files.

Reviewer #2: **Yes: **Mohadeseh Haji Abdolvahab

Reviewer #3: **Yes: **Gustavo J. Vazquez-Zapien

---

## [Editor Report · Acceptance letter]

22 Jun 2023

PONE-D-23-03131R1 

Dataset of single nucleotide polymorphisms of immune-associated genes in patients with SARS-CoV-2 infection 

Dear Dr. Katsaouni:

I'm pleased to inform you that your manuscript has been deemed suitable for publication in PLOS ONE. Congratulations! Your manuscript is now with our production department. 

Kind regards, 

on behalf of

Dr. Asli Suner Karakulah 

Academic Editor

PLOS ONE